# Exposure to heavy metals, bisphenol A, and phthalates: Implications for precocious or delayed puberty

Seung-Ah Choe[ID][1,2☯], Eunjung Kim[3☯], Mina Ha[ID][3]*

1 Department of Preventive Medicine, Korea University College of Medicine, Seoul, Korea, 2 Research and Management Center for Health Risk of Particulate Matter, Korea University, Seoul, Korea, 3 Department of Preventive Medicine, Dankook University College of Medicine, Cheonan, Korea

☯ These authors contributed equally to this work.
* minaha@dku.edu

## Abstract

This study examined the associations between exposure to heavy metals, bisphenol A (BPA), and phthalates and precocious or delayed puberty. This study was a cross-sectional study using the data obtained from the Korea Environmental Exposure and Health Survey in Children and Adolescents. Blood samples were collected to measure lead, mercury, and cadmium levels, whereas spot urine samples were analyzed for BPA, phthalate metabolites, and creatinine levels. Sexual maturation status was assessed using self-measured Tanner stages. Our analyses involved single- and multi-exposure binomial logistic regression models adjusted for age, body mass index, presence of siblings, urban residence, and socioeconomic status. In the multi-exposure models, we applied quantile g-computation mixture analysis to calculate odds ratios (ORs) for precocious and delayed puberty. In the study cohort of 1,424 individuals, precocious puberty was identified in 50 (3.5%) individuals, whereas delayed puberty was identified in 54 (3.8%) individuals. In the single-exposure models, a higher mono-benzyl phthalate concentration was associated with a higher risk of delayed puberty in girls (OR = 2.75, 95% confidence interval: 1.34, 5.66). In the mixture exposure models, exposure to a mixture of BPA and phthalate metabolites was associated with precocious puberty in boys and delayed puberty in both sexes, although the risk estimates were imprecise. Our findings add to the increasing evidence suggesting that exposure to environmental chemicals may contribute to delayed puberty.

## Introduction

Puberty represents a period of important changes leading to sexual maturation and active reproduction function [1]. It encompasses a spectrum of emotional, hormonal, and physical changes, including the growth of pubic hair (pubarche), alterations in

**Data availability statement:** The data are being reviewed and prepared for release, pending institutional approval. Currently data are available from the Environmental Health Research Department of National Institute of Environmental Research (https://www.nier.go.kr, contact point: ojkim21@korea.kr) for researchers who meet the criteria for access to confidential data.

**Funding:** M.H: Ministry of Environment, National Institute of Environmental Research (no grant number), https://www.nier.go.kr S.C: National Research Foundation of Korea (Grant Nos. 2018R1D1A1B07048821 and 2022R1A2C1006364 to SAC), https://www.nrf.re.kr The funders did not play any role in the study design, data collectin and analysis, decision to publish, or preparation of the manuscript.

**Competing interests:** The authors have declared that no competing interests exist.

voice pitch, and an increase in height. It also entails breast development (thelarche) and the onset of menstruation (menarche) in girls, and testicular enlargement in boys [2,3]. On average, puberty onset occurs between the ages of 8 and 13 years in girls and between 9 and 14 years in boys [4]. Early- and late-maturing girls and boys face a heightened risk of experiencing elevated depressive symptoms during early adolescence. Individuals who perceive themselves as maturing later in life are more likely to experience body dissatisfaction. Specifically, an earlier onset of menarche is associated with an increased risk of obesity [5], cardiovascular disease [6], reproductive cancers [7], and overall mortality [8].

Puberty onset in children is a multifaceted process influenced by genetic, environmental, and socioeconomic factors. Concerns have arisen regarding the potential impact of environmental pollutants on pubertal development. Among these pollutants, heavy metals, bisphenol A (BPA), and phthalates have garnered considerable attention because of their widespread use in industrial and consumer products and their known or suspected endocrine-disrupting properties. Research has suggested that prenatal exposure to phthalates and BPA is associated with puberty onset in boys and girls. Specifically, higher levels of these chemicals have been linked to delayed puberty in girls and earlier puberty in boys [9–12].

Given the potential health implications of exposure to heavy metals, BPA, and phthalates on pubertal development, there is a pressing need to investigate this association through comprehensive epidemiological studies. Understanding the impact of environmental pollutants on puberty onset is crucial for identifying modifiable risk factors and informing public health interventions aimed at mitigating the adverse outcomes associated with early or delayed puberty. We hypothesized that higher body burdens of endocrine-disrupting chemicals (EDCs)—specifically heavy metals, BPA, and phthalate metabolites—either singly or in combination, would be associated with altered pubertal timing, manifesting as an increased risk of both precocious and delayed puberty, and that these associations would differ by sex. By leveraging data from a national population-based study, we examined the associations between exposure to heavy metals, BPA, and phthalates and puberty onset, highlighting the complex interplay between environmental factors and pubertal development.

## Methods

### Study population

This study was conducted using the data of Korea Environmental Exposure and Health Survey in Children and Adolescents (KorEHS-C). The survey protocol was performed as previously described [13]. The KorEHS-C is a cross-sectional study based on a nationwide representative sample of 3–18-year-old preschool, elementary-, middle-, and high-school students from 01/03/2012 to 31/10/2014. Among the eligible population, the overall participation rate was 97.5% or 2,388 participants. We excluded those without information on puberty development (n = 529), blood, or urine samples (n = 42). Demographic information was obtained from questionnaires administered to participants and their parents, including questions regarding sociodemographic factors (i.e., household income and each parent's educational

level and occupation) and sexual maturation status (S1 Table). The participants provided blood and urine samples to evaluate their exposure to environmental chemicals and hormones. The study was approved by the Institutional Review Board of Dankook University Hospital, and written informed consent was obtained from the students' parents or guardians before enrollment (DKUH IRB 2012-01-0010, 2012-10-003-001, 2013-05-004-001, and 2014-06-007-002), and the collected data was anonymized and could not be identified individually. The reuse of the data for this study was approved by the Institutional Review Board (DKUH IRB 2025-02-004-001).

## Measurement of environmental chemicals

The methodology for assessing environmental chemicals was conducted as previously described [13]. Blood and urine samples were collected from the participants on the morning of the survey, between 09:00 am and 11:00 am, following an overnight fasting period. Blood samples were collected using metal-free ethylenediaminetetraacetic acid polyethylene tubes (BD, NJ, USA), and urine samples were collected in 15 mL polystyrene conical tubes (SARSTEDT AG&Co., Numbrecht, Germany). Lead, mercury, and cadmium levels were measured using blood samples. BPA, three metabolites of di(2-ethylehexyl) phthalate, mono-2-ethyl-5-carboxypentyl phthalate [MECPP], mono-2-ethyl-5-hydroxyhexyl phthalate [MEHHP], and mono-2-ethyl-5-oxohexyl phthalate [MEOHP], and a metabolite of di-n-butyl phthalate, mono-n-butyl phthalate [MnBP], and a metabolite of benzyl butyl phthalate, mono-benzyl phthalate [MBzP]), and creatinine levels were measured using spot urine samples. These chemicals were chosen based on similar studies [10,14–18].

   The blood samples were stored at –20°C. Owing to the low lead and cadmium levels in the blood, triplicate samples were analyzed. Blood samples for mercury were stored at –70°C until analysis. Mercury analysis was conducted using flow injection cold-vapor atomic absorption spectrometry (AAS) (DMA-80; Milestone, Bergamo, Italy). Initially, the samples were dried in an oxygen stream passing through a quartz tube inside a controlled heating coil. The mercury content was then determined using AAS. Laboratory analyses were performed following standardized quality control procedures. The coefficients of variation were 3.6%, 4.2%, and 3.5% in the reference samples for blood lead, cadmium, and mercury, respectively.

   The urinary BPA levels were measured using high-performance liquid chromatography-tandem mass spectrometry (Agilent 6410 Triple Quad LCMS; Agilent, Santa Clara, CA, USA). Urine aliquots were first adjusted with 30 mL of 2.0 M sodium-acetate buffer (pH 5.0), then fortified with an internal-standard mix containing 100 µg/mL ring-$^{13}C_{12}$-labelled BPA (99%; Cambridge Osotope Laboratories, Andover, MA, USA) and 10 mL of β-glucuronidase/sulfatase (Sigma, St. Louis, MO, USA). Samples were incubated at 37 °C for 3 h to hydrolyze BPA conjugates. After incubation, 100 mL of 2 N HCl was added, the extract was evaporated under nitrogen, and the residue was reconstituted in 1 mL HPLC-grade water inside a 2 mL glass vial. Each analytical batch contained a procedural blank and a quality-control (QC) specimen consisting of pooled urine spiked with 100 ng/mL BPA. Supernatants were cleaned up by solid-phase extraction using Agilent Eclipse, plus C 183.5 mm. Chromatographic separation employed an acetonitrile:water mobile phase (1/4 60:40, v/v) at 0.4 mL/min.

   Samples for measuring the five phthalate metabolites were stored at –20°C. Monoester phthalates were measured using high-performance liquid chromatography-tandem mass spectrometry (Agilent 6410 Triple Quad LCMS; Agilent, Santa Clara, CA, USA). The between-day coefficient of variation for the assay ranged from 0.5% to 8.9%. For undetected environmental chemicals, the value was replaced with half of the detection limit. The undetected levels ranged from 0% to 6.8% (S2 Table).

## Identification of precocious and delayed puberty

Sexual maturation status was assessed based on self-measured Tanner stages [19]. These self-measured stages of puberty can be used as a valid indicator of sexual development in community-based studies in which physical examination is not feasible [19,20]. Precocious puberty is defined as the appearance of any Tanner stage 2 secondary sexual

characteristics before the age of 8 years or menarche before the age of 10 years in girls. For boys, self-reported Tanner stage 2 before the age of 9 years was considered indicative of precocious puberty [21]. Delayed puberty was identified if girls have not reached Tanner stage 2 or experienced menarche by the age of 13 years or if boys have not reached Tanner stage 2 by the age of 14 years [22]. This cross-sectional survey focused on the status of sexual development at the time of assessment. While girls aged 9–13 years were eligible for precocious puberty identification, determining precocious or delayed puberty for boys in this age group was not possible based on the single measurement taken at the time of the survey. Consequently, we could only determine precocious puberty for boys aged ≤8 years and delayed puberty for girls aged ≥13 years and boys aged ≥14 years. After further excluding girls aged < 8 years and boys aged 9–13 years (n = 393), the final study population comprised 1,424 participants (874 girls and 550 boys).

## Covariates

We selected covariates based on a directed acyclic graph and data availability (S3 Fig). The selected covariates included participants' age at the time of the tests, body mass index (BMI), presence of siblings, urban residence, and socioeconomic status (SES). Age was included as a covariate because it can act as a common cause influencing both chemical exposure and pubertal status [23]. Having siblings may reflect household factors that indirectly influence both exposure and developmental outcomes [24]. We identified those who lived in urban areas (metropolitan or city) based on participants' home addresses. SES was defined by the higher educational attainment of either parent, and household income exceeding the 2014 median. Parental occupation was excluded from the list of covariates because the non-response rate exceeded 20%.

## Statistical analysis

Descriptive statistics summarized the baseline clinical characteristics of the study sample. Due to their skewness, the levels of all the measured environmental chemicals were log-transformed in regression models. The Spearman correlation test was used to evaluate pairwise associations among the environmental chemicals and to identify strongly correlated compounds.

Single-exposure models were used to explore the pairwise association between each chemical and abnormal puberty onset, with separate analyses conducted for girls and boys within the age ranges at risk. Each single-exposure model adjusted for age, BMI, presence of siblings, urban residence, and SES. We calculated the odds ratios (ORs) for precocious and delayed puberty per interquartile range increase in the log-transformed level of each chemical.

In the multi-exposure models, we applied quantile g-computation (QGC) mixture analysis to investigate the association of exposure to mixtures of environmental chemicals, adjusting for all covariates listed in the single-exposure models. The QGC approach integrates elements from the weighted quantile sum with the causal inference technique known as g-computation [25]. This methodology is especially beneficial for evaluating the collective impact of multiple environmental factors on health outcomes, with the capacity to reveal nonlinear associations. Compared to most traditional multivariable regression models that suffer from collinearity issues when including multiple correlated variables, QGC method can reduce the impact of extreme correlations between individual exposures by working with quantiles of mixtures rather than the original continuous values [25]. We divided the exposure into two groups: three heavy metals and six chemicals (BPA and five phthalate metabolites). The OR of each chemical mixture associated with a simultaneous one-quartile increase in the mean log-transformed exposure level was estimated using the *qgcomp* package in R. We provided weights indicating the contribution of the individual components of the mixture to the overall estimate. We further examined a potential interaction by heavy metal exposure in the association between chemical mixture and abnormal puberty. Stratified associations between mixture exposure to phthalate metabolites and abnormal puberty by low (less than median) or high (median or higher) level were assessed for heavy metals which show positive association with abnormal puberty. R (version 4.2.3) was used for all statistical analyses and plots.

## Results

Precocious puberty was identified in 50 (3.5%) participants, and delayed puberty in 54 (3.8%) participants (S1 Table). In both sexes, mean BMI and the proportion with siblings were higher in the delayed-puberty analysis cohort (Table 1). The precocious-puberty cohort had a higher percentage of participants with at least one parent holding a college or university degree. Blood and urinary chemical concentration ranges largely overlapped between the two cohorts. Because the delayed-puberty cohort was older on average, Tanner stages were correspondingly higher.

In the study population, precocious puberty was more frequent among boys (12 of 166, 7.2%) than girls (38 of 874, 4.3%). Similarly, delayed puberty was observed in 10.9% of boys (42 of 384) and 3.2% of girls (12 of 378). The proportion of participants living in rural areas was higher among those with delayed puberty. The mean levels of cadmium, BPA, and all phthalate metabolites were generally higher in participants with precocious puberty than in those with delayed or normal puberty.

The mean cadmium, MECPP, MnBP, MEOHP, and MEHHP levels were higher in the order of delayed, normal, and precocious puberty. Mercury levels did not differ significantly among the three groups. A strong correlation was observed between MECPP and MEOHP ($rho = 0.94$), MECPP and MEHHP ($rho = 0.92$), and MEOHP and MEHHP ($rho = 0.97$; S4 Fig.). For the other pairs, the correlation coefficients ranged from 0.01 to 0.66.

In the single-exposure models, the association between the nine measured chemicals and precocious puberty was generally inconsistent across both sexes. Moreover, the confidence intervals (CIs) of the estimates were imprecise when controlling for age, BMI, presence of siblings, urban residence, and SES in girls and boys (Table 2). Among girls, a higher MBzP concentration was associated with a higher risk of delayed puberty (OR = 2.75, 95% CI: 1.34, 5.66). For the other chemicals, the association with delayed puberty did not show significant associations.

None of the ORs for the mixture exposure reached statistical significance. The association between the mixture of BPA and phthalate metabolites and precocious puberty was positive in boys (OR = 2.34, 95% CI: 0.36, 15.14; Table 3). The association of heavy metals or a mixture of BPA and phthalate metabolites with precocious puberty was minimal. For delayed puberty, exposure to a mixture of heavy metals (OR = 1.35, 95% CI: 0.80, 2.29) or BPA and phthalate metabolites (1.29, 95% CI: 0.73, 2.31) showed a positive association among boys, whereas the associations with heavy metals, BPA, and phthalate mixture were generally inconsistent among girls.

For precocious puberty, lead (98.7% for girls and 100.0% for boys) contributed positively to the overall mixture effects of heavy metals, and BPA and phthalate metabolites, respectively. For delayed puberty, however, it was the primary negative contributor to the mixture effect of heavy metals in girls (61.9%), but the primary positive contributor in boys (79.6%).

MBzP contributed positively to the overall mixture effect on both abnormal timing of puberty: 32.2% and 25.4% for precocious puberty, and 52.7% and 14.9% for delayed puberty, among girls and boys, respectively. MnBP positively (in boys) or negatively (in girls) contributed to both precocious and delayed puberty. MECCP contributed positively to the mixture effect on precocious puberty in girls and delayed puberty in boys, but negatively to the delayed puberty in girls and precocious puberty in boys. The contribution directions of BPA or other phthalate metabolites are complex by different abnormal puberty or by sex. Stratified associations between mixture exposure to BPA, and phthalate metabolites and abnormal puberty by urinary cadmium level showed similar findings, implicating no significant interaction by the level of heavy metals (S5 Fig).

## Discussion

Our findings reveal a complex relationship between exposure to environmental chemicals and abnormal puberty onset. While the levels of the nine measured chemicals were higher in the order of precocious, normal, and delayed pubertal groups, weak associations were observed between precocious puberty and individual chemicals. A positive association with delayed puberty was observed only for MBzP and solely among girls. Although boys showed an increased risk of precocious and delayed puberty in associations with both mixtures of heavy metals or BPA and phthalate metabolites, the

**Table 1. Participants characteristics of cohorts for analyzing precocious or delayed puberty stratified by sex in the Korea Environmental Exposure and Health Survey in Children and Adolescents (n = 1,424).**

| Variables and chemical levels | Girls | | Boys | |
|---|---|---|---|---|
| | Precocious puberty cohort (all) | Delayed puberty cohort (age ≥ 13) | Precocious puberty cohort (age < 9) | Delayed puberty cohort (age ≥ 14) |
| N | 874 | 378 | 166 | 384 |
| Age, years (range) | 12.19 [9.63, 15.15] | 15.73 [14.17, 16.91] | 7.74 [7.36, 8.38] | 16.30 [15.24, 17.37] |
| Body mass index, kg/m² | 19.36 (3.31) | 21.12 (3.02) | 16.96 (2.57) | 22.15 (3.85) |
| Living in urban area | 304 (34.8) | 126 (33.3) | 44 (26.6) | 138 (35.9) |
| Presence of sibling | 375 (43.0) | 187 (49.5) | 62 (37.3) | 197 (51.4) |
| Parents with college or university degrees | 459 (54.1) | 173 (47.1) | 100 (61.3) | 162 (44.0) |
| *Heavy metals*[c] | | | | |
| Blood lead, ug/dL | 1.87 [1.44, 2.44] | 1.84 [1.36, 2.42] | 1.75 [1.39, 2.15] | 1.87 [1.41, 2.60] |
| Blood mercury, ug/L | 1.10 [0.87, 1.39] | 0.99 [0.76, 1.23] | 1.38 [1.11, 1.63] | 1.22 [0.93, 1.48] |
| Urinary cadmium, ug/L $_{creatinine}$ | 0.29 [0.21, 0.38] | 0.25 [0.19, 0.33] | 0.35 [0.27, 0.44] | 0.21 [0.15, 0.29] |
| *Urinary BPA and phthalates metabolites*[c] | | | | |
| BPA, ug/g $_{creatinine}$ | 1.10 [0.58, 2.12] | 0.76 [0.41, 1.52] | 1.85 [0.68, 3.81] | 0.77 [0.46, 1.39] |
| MBzP, ug/g $_{creatinine}$ | 4.92 [2.63, 10.86] | 3.71 [2.00, 7.73] | 8.71 [3.61, 15.82] | 4.01 [2.16, 8.21] |
| MECPP, ug/g $_{creatinine}$ | 35.84 [21.66, 58.29] | 23.64 [15.57, 34.92] | 68.03 [51.42, 102.69] | 22.14 [15.57, 31.80] |
| MnBP, ug/g $_{creatinine}$ | 44.67 [29.95, 72.62] | 32.09 [23.48, 43.39] | 78.00 [52.55, 122.77] | 29.70 [21.17, 40.29] |
| MEOHP, ug/g $_{creatinine}$ | 17.65 [10.80, 29.90] | 11.70 [8.19, 17.07] | 37.27 [27.43, 56.87] | 11.34 [7.86, 15.95] |
| MEHHP, ug/g $_{creatinine}$ | 24.05 [14.40, 41.23] | 16.40 [10.25, 24.72] | 51.58 [35.10, 75.49] | 16.39 [11.01, 23.76] |
| *Tanner stage* | | | | |
| *Pubic hair* | | | | |
| I | 362 (42.3) | 2 (0.5) | 133 (82.1) | 2 (0.5) |
| II | 162 (18.9) | 78 (20.8) | 18 (11.1) | 16 (4.2) |
| III | 164 (19.2) | 140 (37.3) | 9 (5.6) | 63 (16.7) |
| IV | 142 (16.6) | 130 (34.7) | 1 (0.6) | 137 (36.2) |
| V | 25 (2.9) | 25 (6.7) | 1 (0.6) | 160 (42.3) |
| Breast/Testicular volume | | | | |
| I | 186 (21.8) | 1 (0.3) | 93 (56.0) | 28 (7.3) |
| II | 155 (18.1) | 10 (2.7) | 50 (30.1) | 69 (18.0) |
| III | 238 (27.8) | 128 (34.1) | 13 (7.8) | 133 (34.6) |
| IV | 191 (22.3) | 156 (41.6) | 6 (3.6) | 106 (27.6) |
| V | 85 (9.9) | 80 (21.3) | 0 (0.0) | 56 (14.6) |

Urinary levels of cadmium, BPA, and phthalate metabolites were creatinine-adjusted.

[a]P-values for differences among the three groups.

[b]The median household income for 2014 was used to determine whether income was above the median.

[c]The values represent median and interquartile range.

Abbreviations: SD, standard deviation; BPA, bisphenol A; MBzP, mono-benzyl phthalate; MECPP, mono-2-ethyl-5-carboxypentyl phthalate; MnBP, mono-N-butyl phthalate; MEOHP, mono-2-ethyl-5-oxohexyl phthalate; MEHHP, mono-2-ethyl-5-hydroxyhexyl phthalate.

risk estimates were imprecise. These findings offer insights into the potential roles of heavy metals, BPA, and phthalate metabolites in pubertal timing in children and adolescents.

Previous studies have yielded conflicting results regarding the link between exposure to various endocrine-disrupting chemicals and puberty onset. For instance, low-level exposure to lead and cadmium has been inversely associated with inhibin B levels, suggesting that these metals may play a role in delaying puberty onset in girls [26]. By employing a

**Table 2. Association between levels of heavy metals, BPA, and phthalate metabolites and puberty onset in the single-exposure models in the Korea Environmental Exposure and Health Survey in Children and Adolescents (n = 1,424).**

| Exposure | Risk of precocious puberty | | Risk of delayed puberty | |
|---|---|---|---|---|
| | OR (95% CI) | | OR (95% CI) | |
| | Girls | Boys | Girls | Boys |
| *Log-transformed blood and urinary heavy metals* | | | | |
| Lead, ug/dL | 1.54 (0.60, 4.00) | 0.27 (0.03, 2.18) | 0.94 (0.20, 4.45) | 1.50 (0.68, 3.33) |
| Mercury, ug/L | 1.16 (0.39, 3.48) | 3.28 (0.47, 23.04) | 0.42 (0.07, 2.37) | 2.48 (0.93, 6.56) |
| Cadmium, ug/L $_{creatinine}$ | 1.23 (0.57, 2.66) | 0.64 (0.13, 3.02) | 0.64 (0.15, 2.67) | 0.91 (0.40, 2.06) |
| *Log-transformed urinary levels of BPA and phthalate metabolites* | | | | |
| BPA, ug/g $_{creatinine}$ | 1.26 (0.87, 1.82) | 1.18 (0.65, 2.14) | 1.35 (0.72, 2.55) | 1.02 (0.69, 1.50) |
| MBzP, ug/g $_{creatinine}$ | 1.16 (0.82, 1.64) | 1.63 (0.92, 2.90) | 2.75 (1.34, 5.66) | 1.13 (0.78, 1.64) |
| MECPP, ug/g $_{creatinine}$ | 1.23 (0.66, 2.30) | 0.55 (0.17, 1.78) | 1.12 (0.32, 3.93) | 1.68 (0.87, 3.24) |
| MnBP, ug/g $_{creatinine}$ | 0.68 (0.36, 1.31) | 1.63 (0.52, 5.09) | 0.86 (0.23, 3.28) | 1.32 (0.65, 2.68) |
| MEOHP, ug/g $_{creatinine}$ | 1.10 (0.65, 1.86) | 0.88 (0.40, 1.93) | 1.37 (0.43, 4.38) | 1.48 (0.78, 2.79) |
| MEHHP, ug/g $_{creatinine}$ | 1.11 (0.66, 1.84) | 0.89 (0.42, 1.90) | 1.00 (0.33, 3.04) | 1.36 (0.75, 2.47) |

ORs of precocious and delayed puberty per interquartile range increased in the log-transformed level of each chemical in the multiple logistic regression models adjusted for age, body mass index, presence of siblings, urban residence, and socioeconomic status.

Abbreviations: OR, odds ratio; CI, confidence interval; BPA, bisphenol A; MBzP, mono-benzyl phthalate; MECPP, mono-2-ethyl-5-carboxypentyl phthalate; MEHHP, mono-2-ethyl-5-hydroxyhexyl phthalate; MEOHP; mono-2-ethyl-5-oxohexyl phthalate; MnBP, mono-N-butyl phthalate.

cross-sectional design, some studies have replicated the positive association between blood lead and cadmium levels and delayed menarche [15,16,27], whereas others have found that lead and mercury levels are associated with earlier menarche [27,28]. Although the literature on this topic is limited, studies have reported a positive association between heavy metal exposure and delayed puberty in boys [18,29]. This inconsistency may stem from variations in the exposure matrices (blood, urine, or hair) and the definitions used to determine puberty [14,30]. Additionally, our study population comprised children younger than those in previous studies.

Specifically, our mixture analyses revealed that exposure to lead predominantly drove the positive association between exposure to a mixture of heavy metals and delayed puberty in boys. This finding supports previous observations of a positive association between heavy metals and delayed puberty. Lead absorbs readily into the bloodstream and accumulates in bone, where it can be remobilized during growth spurts [31]. Because children's bones are actively remodeling, even moderate lead exposures can translate into sustained internal doses that interfere with developmental processes. Experimental and observational data show that lead can perturb the hypothalamic–pituitary–gonadal (HPG) axis. In animal models, lead exposure delays the rise in luteinizing hormone and slows ovarian follicle maturation; in humans, higher blood lead levels have been associated with later pubertal development in girls [32].

Evidence regarding the effects of BPA exposure and phthalate metabolites during the peripubertal period has been inconclusive. A systematic review reported a positive association between postnatal BPA exposure and earlier thelarche and late pubarche in girls [17]. However, in the same study, postnatal phthalate exposure was not associated with testicular enlargement in boys. In a longitudinal cohort study of Russian boys, MBzP was associated with a slower trajectory of pubertal progression. MBzP is a metabolite of antiandrogenic BzBP, which decreases testosterone production and causes alterations in androgen-organized tissues in a dose-additive manner [33]. Our observation of MBzP as a significant contributor to the association with precocious and delayed puberty was partially aligned with previous findings. An in vivo study found that maternal exposure to butyl benzyl phthalate (the parent compound of MBzP) increased the incidence of undescended testes in rat fetuses [34]. Despite these observations, the association between heavy metals or a mixture of BPA and phthalate metabolites and precocious or delayed puberty appeared minimal in our study cohort. This

**Table 3. Association between levels of chemical mixtures and precocious or delayed puberty stratified by sex in the Korea Environmental Exposure and Health Survey in Children and Adolescents (n = 1,424).**

| Variables | Girls | | | Variables | Boys | | |
|---|---|---|---|---|---|---|---|
| | Contribution to positive/ negative effect | Positive/negative coefficient | Overall mixture effect OR (95% CI) | | Contribution to positive/negative effect | Positive/negative coefficient | Overall mixture effect OR (95% CI) |
| Precocious puberty | | | | | | | |
| *Heavy metals* | | | | *Heavy metals* | | | |
| Blood lead | 98.7% | 0.097 | 1.06 (0.62, 1.81) | Blood lead | 100.0% | 1.080 | 1.29 (0.24, 7.00) |
| Urinary cadmium | 1.3% | | | Blood mercury | 78.9% | −0.828 | |
| Blood mercury | 100.0% | −0.038 | | Urinary cadmium | 21.1% | | |
| *Urinary BPA and phthalate metabolites* | | | 1.10 (0.62, 1.95) | *Urinary levels of BPA and phthalate metabolites* | | | 2.34 (0.36, 15.14) |
| MBzP | 32.2% | 0.609 | | MBzP | 25.4% | 3.520 | |
| MECPP | 32.9% | | | MnBP | 72.0% | | |
| BPA | 25.9% | | | BPA | 2.6% | | |
| MEHHP | 9.0% | −0.514 | | MECPP | 62.8% | −2.670 | |
| MnBP | 66.5% | | | MEHHP | 5.4% | | |
| MEOHP | 33.5% | | | MEOHP | 31.9% | | |
| Delayed puberty | | | | | | | |
| *Heavy metals* | | | | *Heavy metals* | | | |
| Blood lead | 61.9% | −0.731 | 0.48 (0.16, 1.45) | Blood lead | 79.6% | 0.454 | 1.35 (0.80, 2.29) |
| Blood mercury | 21.3% | | | Blood mercury | 20.4% | | |
| Urinary cadmium | 16.8% | | | Urinary cadmium | 100.0% | −0.151 | |
| *Urinary levels of BPA and phthalate metabolites* | | | 1.33 (0.35, 5.05) | *Urinary levels of BPA and phthalate metabolites* | | | 1.29 (0.73, 2.31) |
| MBzP | 52.7% | 2.380 | | MECPP | 64.8% | 0.712 | |
| MEOHP | 37.1% | | | MnBP | 20.3% | | |
| BPA | 10.2% | | | MBzP | 14.9% | | |
| MnBP | 35.9% | −2.100 | | MEHHP | 41.2% | −0.455 | |
| MECPP | 35.2% | | | BPA | 36.9% | | |
| MEHHP | 28.9% | | | MEOHP | 21.9% | | |

Quantile g-computation was used to investigate the effects of a chemical mixture of heavy metals, BPA, and phthalate metabolites in a multiple logistic regression model adjusted for age, body mass index, presence of siblings, urban residence, and socioeconomic status. The OR of each chemical mixture for a simultaneous one-quartile change in average chemical exposure was estimated. The contribution of each chemical is presented with the weight (%).

Abbreviations: OR, odds ratio; CI, confidence interval; BPA, bisphenol A; MBzP, mono-benzyl phthalate; MECPP, mono-2-ethyl-5-carboxypentyl phthalate; MEHHP, mono-2-ethyl-5-hydroxyhexyl phthalate; MEOHP; mono-2-ethyl-5-oxohexyl phthalate; MnBP, mono-N-butyl phthalate.

finding suggests that the mechanisms underlying the effects of environmental chemical exposure on pubertal timing may differ in the context of joint or concurrent exposure. For example, the positive association between phthalate exposure and delayed puberty was stronger in obese boys [35,36]. This may explain the weaker association among the boys of our study population who show generally low BMI. By quantifying joint exposures, our approach enhances comparability with prior small or single-chemical studies.

Additionally, participants living in rural areas were more likely to experience delayed puberty. Previous studies have reported that the risk of early menarche increases with a higher regional deprivation index and urbanization [24].

Urbanization is associated with a higher risk of childhood overweight and obesity, which can lead to early puberty [37,38]. Children living in rural areas are less likely to be exposed to hazardous environments, including air pollution and contaminated water, which are associated with precocious puberty [39]. Unfavorable environmental features of urban areas such as limited recreational facilities, poor safety, and high pollution were consistently associated with early menarche among girls [40]. Given the higher risk of early puberty in urban areas, the proportion of individuals with delayed puberty would have been relatively higher in rural areas.

This study has several limitations that warrant consideration. First, the sample size might have limited our ability to detect associations with precocious puberty, given its rarity. Assuming a one-quartile increase in exposure confers a 1.5-fold higher risk of precocious puberty, our observed incidence yields only about 60% power. Moreover, with few events relative to the number of exposure variables, there is an elevated risk of model overfitting, imprecision of effect estimates and reduced robustness of the findings. Reliance on a single dataset may magnify these issues, underscoring the need for larger samples and independent replication in future studies to ensure robustness. Nevertheless, as an exploratory study of environmental chemical mixtures, our results can guide future sample-size calculations needed to robustly detect such effects. Pooled or longitudinal study including a large sample would be required for confirmatory inference on multi-exposure associations. Second, only nine chemicals were measured; Unmeasured environmental, nutritional, or genetic factors may remain as potential sources of residual confounding effect. Third, this study did not capture the timing or duration of exposure to environmental chemicals throughout the critical periods of pubertal development, particularly concerning heavy metals with long metabolic half-lives. This might have affected our results and needs to be considered for balanced interpretation of our findings. Fourth, the assessment of pubertal timing relies on a combination of self-reporting and physical examination, which introduces the potential for measurement errors or misclassification of pubertal stages, especially in younger participants. However, given the prevalence of abnormal puberty timing is comparable to prior studies, we believe the potential measurement bias would have been minimal. In addition, our measurement of precocious puberty lacks the detailed timing information needed to support a dose–response analysis. Lastly, although the KorEHS-C data are a representative sample of Korean children and adolescents aged 6–18 years, excluding boys aged 9–13 years in the present study undermines this representativeness. This aspect may limit the generalizability of our findings to a broader population of children and adolescents. The imprecision of risk estimates, particularly for associations with precocious puberty, underscores the need for caution in interpreting these results. Although the risk estimates were imprecise, effect sizes among girls were generally small, suggesting any association between chemical mixtures and abnormal puberty in this group is likely negligible. In boys, despite wide confidence intervals, the consistently positive direction of the estimates indicates a possible link between chemical exposure and abnormal puberty. Given the limited research on the association between multiple chemical exposures and early puberty, our exploratory analysis may serve as a starting point to stimulate further investigation. Finally, the cross-sectional design of this study restricts our ability to establish temporal relationships or infer causality between environmental chemical exposure and pubertal timing. Nonetheless, by including boys and girls and examining precocious and delayed puberty, our analysis offers a comprehensive understanding of how environmental chemical exposure affects pubertal timing across sexes and the spectrum of abnormal puberty. Moreover, we conducted single- and joint-exposure analyses to provide new evidence on the role of chemicals in delayed puberty, an area that has been less studied in environmental epidemiology.

## Conclusions

Our findings contribute to the growing body of evidence implicating the potential role of heavy metals, BPA and phthalate metabolites in pubertal development. Further longitudinal studies with larger sample sizes are needed to elucidate causal relationships and underlying mechanisms driving these associations.

## Supporting information

**S1 Table. Characteristics of study participants of the KorEHS-C (n = 1,424).**
(PDF)

**S2 Table. Number of samples below the limit of detection of measurement.**
(PDF)

**S3 Fig. Directed acyclic graph for the causal pathway.**
(PDF)

**S4 Fig. Pairwise correlation structure of all measured chemicals.**
(PDF)

**S5 Fig. Stratified association between mixture exposure to BPA, and phthalate metabolites and abnormal puberty.**
(PDF)

## Acknowledgments

We thank all the participating students, their parents or guardians, and the schoolteachers for their time and kind help in the pilot study.

## Author contributions

**Conceptualization:** Seung-Ah Choe, Eunjung Kim, Mina Ha.

**Data curation:** Eunjung Kim.

**Formal analysis:** Seung-Ah Choe, Eunjung Kim.

**Funding acquisition:** Seung-Ah Choe, Mina Ha.

**Investigation:** Mina Ha.

**Methodology:** Seung-Ah Choe.

**Resources:** Eunjung Kim.

**Supervision:** Mina Ha.

**Validation:** Seung-Ah Choe, Eunjung Kim.

**Writing – original draft:** Seung-Ah Choe, Eunjung Kim.

**Writing – review & editing:** Seung-Ah Choe, Eunjung Kim, Mina Ha.

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
