## [Decision Letter · Decision Letter 0]

23 Apr 2025

Dear Dr. Ha,

Thank you for submitting your manuscript to PLOS ONE. After careful consideration, we feel that it has merit but does not fully meet PLOS ONE’s publication criteria as it currently stands. Therefore, we invite you to submit a revised version of the manuscript that addresses the points raised during the review process.

We look forward to receiving your revised manuscript.

Kind regards,

Li Yang, M.D.

Academic Editor

PLOS ONE

Journal Requirements:

2. In the online submission form you indicate that your data is not available for proprietary reasons and have provided a contact point for accessing this data. Please note that your current contact point is a co-author on this manuscript. According to our Data Policy, the contact point must not be an author on the manuscript and must be an institutional contact, ideally not an individual. Please revise your data statement to a non-author institutional point of contact, such as a data access or ethics committee, and send this to us via return email. Please also include contact information for the third party organization, and please include the full citation of where the data can be found.

Reviewers' comments:

Reviewer's Responses to Questions

**Comments to the Author**

1. Is the manuscript technically sound, and do the data support the conclusions?

Reviewer #1: Partly

Reviewer #2: No

Reviewer #3: Partly

2. Has the statistical analysis been performed appropriately and rigorously?

Reviewer #1: Yes

Reviewer #2: No

Reviewer #3: No

3. Have the authors made all data underlying the findings in their manuscript fully available?

Reviewer #1: Yes

Reviewer #2: No

Reviewer #3: No

4. Is the manuscript presented in an intelligible fashion and written in standard English?

Reviewer #1: Yes

Reviewer #2: No

Reviewer #3: Yes

Reviewer #1: The manuscript titled 'Exposure to heavy metals, bisphenol A, and phthalates: Implications for precocious or delayed puberty' is interesting and written well. I have only a few remarks:

Comment 1: In introduction, the author needs to put forward the hypothesis of this study and clarify the content and significance of his own research. In addition, What are the differences or innovations between this study and these findings?

Comment 2: In line 128-129, the authors need to add detailed test procedures or add references.

Comment 3: In table 1, need to label units, especially in Blood and urinary measurement of heavy metals, like table 2.

Comment 4: In results, it is suggested to add the corresponding figures, here are all tables. In addition, is there any correlation analysis that can be done between these data?

Comment 5: in discussion, it is suggested that the authors further enhance the depth of the discussion section of this paper.

Reviewer #2: The manuscript has several major shortcomings that impact its overall scientific rigor. The abstract (lines 38–54) does not clearly highlight the cross-sectional nature of the study, which significantly limits any causal interpretation between exposure to environmental chemicals and puberty outcomes. The reliance on self-assessed Tanner stages (line 43) introduces potential bias due to misclassification, especially among younger participants who may lack the ability to accurately evaluate their developmental stage. Although the study employs mixture models (lines 45–46), the results are frequently described as imprecise (lines 52–53, 260), suggesting that the findings may lack robustness and could be due to chance. Among the nine chemicals measured, only mono-benzyl phthalate (MBzP) showed a statistically significant association with delayed puberty in girls (lines 49–51, 259), while associations for other chemicals were weak and inconsistent, particularly in boys. The analysis lacks a detailed discussion of dose-response relationships, which is critical for establishing the biological plausibility of the reported associations. The paper reviews inconsistent findings from previous literature (lines 265–275) but does not adequately explain how the current study design resolves these inconsistencies or adds clarity. The identification of lead as a key driver of delayed puberty in mixture analysis (lines 276–278) is not thoroughly validated by sex-specific or stratified analyses, weakening the strength of the conclusion. The number of identified cases of precocious and delayed puberty is relatively small (lines 47–48), raising concerns about the statistical power of the study and the reliability of the regression models used. These issues collectively limit the interpretability and generalizability of the study’s findings.

Reviewer #3: Here is a list of specific comments. Note: line and page numbering in reviews and comments is based on

ruler applied in Editorial Manager-generated PDF.

1. Page 3, line 48: There might be issues due to the small number of events per variable in the models, especially the multi-exposure models. The first limitation in the Discussion section addressed the low power issue. I suggest expanding the limitation to other issues such as overfitting, biased estimation, model instability, etc.

2. Page 5, lines 91–92: I suggest reporting the number of participants excluded due to missing information on pubertal development, missing blood samples, and missing urine samples separately.

3. Page 7, lines 135–145: It might be worth considering two separate cohorts, one for the precocious puberty and the other for the delayed puberty. The precocious puberty cohort would consist of girls with age 10 years or younger and boys with age 8 or younger. The delayed puberty cohort would consist of girls with age 13 or older and boys with age 14 or older. In other words, Table S1 would be split into two tables.

4. Page 7, lines 148–150: I suggest rewriting the sentence as follows. ‘The selected covariates included participants’ age at the time of the tests, body mass index (BMI), presence of siblings, urban residence, and socioeconomic status (SES).’

5. Page 7, line 150: BMI is a well-known metric and does not require a definition. If you would like to keep the sentence, I suggest revising the sentence as ‘BMI was calculated as weight in kilograms divided by height in meters squared.’

6. Page 7, lines 153–154: I suggest revising “the median” as ‘the median household income for 2014’ and removing the subsequent sentence, “the median household income for 2014 was . . . .” I suggest being specific about the median household income for 2014.

7. Page 7, lines 157–158: I suggest rewriting the sentence as follows. ‘Descriptive statistics summarized the baseline clinical characteristics of the study sample.’

8. Page 8, lines 158–159: I suggest rewriting the sentence as follows. ‘Due to their skewness, the urine levels of the measured environmental chemicals were log-transformed in regression models.’

9. Page 8, lines 159–160: I suggest stating the purpose of examining the correlations among the environmental chemicals.

10. Page 8, lines 161–162: I suggest revising the sentence as ‘each single-exposure model adjusted for age, BMI, presence of siblings, urban residence, and SES,’ and relocating the revised sentence to line 165 (after the next sentence).

11. Page 8, line 163: I suggest unifying the use of “environmental chemicals” and “heavy metals, BPA, or phthalate metabolites.”

12. Page 8, line 166: The two blood chemicals, Pb and Hg, were not log-transformed. The phrase was not applicable to all chemicals.

13. Page 8, line 169: I suggest revising “all covariates” as ‘all covariates listed in the single-exposure models.’

14. Page 8, lines 174–175: Please clarify if it was the one-quartile change in average chemical exposure or in average ‘log’ chemical exposure because the urine levels were log-transformed.

15. Page 8, line 180: There was no Table S3. Did you mean Table S1?

16. Table 1: I suggest not reporting characteristics by the outcome status. I suggest replacing the current Table 1 with Table S1 (after revision).

17. Figure 1: For the same reason as Comment 16 above, I suggest removing Figure 1.

18. Page 10, lines 209–210: Following Comment 9 above, what was the implication of these observed high correlations? If the purpose of reporting correlations was to avoid collinearity, these high correlations might distort the estimates of the multi-exposure models.

19. Table 2: I suggest providing a better indication of which exposures were log-transformed.

20. Table S1: Different age categories in girls and boys made the distributions difficult to interpret. Please consider the two-cohort approach suggested in Comment 3 above.

21. Figure S3: I suggest clarifying what EDC stands for.

**Do you want your identity to be public for this peer review?** For information about this choice, including consent withdrawal, please see our Privacy Policy

Reviewer #1: No

Reviewer #2: No

Reviewer #3: No

---

## [Author Response · Author response to Decision Letter 1]

4 Jun 2025

We revised the section of data and code availability according the Journal Policy.

We also responded to each point of comments of editor and all reviewers. Please find the uploaded response letter.

---

## [Decision Letter · Decision Letter 1]

29 Jun 2025

Dear Dr. Ha,

Thank you for submitting your manuscript to PLOS ONE. After careful consideration, we feel that it has merit but does not fully meet PLOS ONE’s publication criteria as it currently stands. Therefore, we invite you to submit a revised version of the manuscript that addresses the points raised during the review process.

**Major concerns:**

1. Statistical power and model reliability: The number of cases of precocious puberty (50 cases) and delayed puberty (54 cases) is too low. Especially, the exposure model includes 9 chemicals, which may lead to overfitting of the model and unstable estimates (e.g., OR for boys with precocious puberty = 2.34, 95% CI: 0.36 - 15.14). The authors mentioned that the test power is only about 60% (assuming OR = 1.5), but did not fully discuss its potential impact on negative results (such as the possibility that no association with other chemicals may be due to insufficient power). I think the discussion section needs to be expanded to clearly explain the impact of small sample size on the mixed model (such as the risk of overfitting) and suggest the required sample size for future studies. Add sensitivity analysis and add model stability tests (such as Bootstrap method to verify the confidence interval of OR).

2. Limitations of exposure assessment: Relying solely on a single biological sample (blood/urine) cannot reflect the dynamic of long-term exposure, especially for heavy metals with long half-lives (such as lead accumulation in bones). Other environmental pollutants (such as pesticides, flame retardants) were not included, which may miss important confounding factors. The authors need to strengthen the discussion and clearly state the limitations of single measurement in the "Limitations" section. It is recommended that future studies adopt a longitudinal design or cumulative exposure indicators. Supplementary analysis: If the data permit, analyze the interactions between chemical substances (such as heavy metals and phthalates).

3. Rigor of result interpretation: Only MBzP showed a significant effect on girls' delayed puberty (OR = 2.75), while the other associations were not significant or had wide confidence intervals. However, the conclusion that "chemical mixtures are associated with delayed puberty" may be an overinterpretation. The discussion on the mechanism of urban-rural differences (with a higher proportion of girls experiencing delayed puberty in rural areas) is insufficient (such as whether it is related to pollutant exposure levels or social economic factors?). Please revise the conclusion statement to avoid suggesting a "wide correlation of mixtures", and focus on the significance of MBzP and the potential role of lead in boys. Delve deeper into the differences between urban and rural areas, and supplement with literature support (such as citing Oh et al. 2024's discussion on air pollution and precocious puberty). Distinguish the effects of pollution exposure from those of confounding factors such as obesity.

4. Methodological details and data accuracy: In Table 3, "MBaP" should be "MBzP" (Page 21). Inconsistent ethical approval numbers: The main text states 2013-05-004-001 (Page 12), but the supplementary material is 2013-05-004_(III).pdf (Page 38). The correlation coefficients are extremely high (such as MEOHP-MEHHP, ρ=0.97) and the impact of multicollinearity on the model has not been discussed. The authors should correct the data errors, unify the ethical approval numbers, and correct "MBaP" in Table 3. Provide an explanation for handling multicollinearity, clearly explain how QGC reduces the influence of multicollinearity (such as quantile transformation), or provide a VIF analysis in the supplementary material.

**Minor concerns:**

1. Chart optimization: Table 1: Add units (such as blood lead μg/dL) and provide a footnote stating "median [IQR]" or "mean (SD)". Figure S4: Simplify the correlation coefficient graph, highlighting the high correlations with |ρ| > 0.7.

2. Discussion Extension: Analyze the potential mechanisms underlying the significant effect of MBzP in girls and the lack of such effect in boys (such as differences in sex hormone pathways). Refer to the latest literature to support the discussion on the urban-rural disparity.

3. Terminology Consistency: The term "phthalate metabolites" (instead of "phthalates") is uniformly used throughout the text to avoid confusion between the parent compound and its metabolites.

We look forward to receiving your revised manuscript.

Kind regards,

Li Yang, M.D.

Academic Editor

PLOS ONE

**Additional Editor Comments:**

Please further address editor comments.

Reviewers' comments:

Reviewer's Responses to Questions

**Comments to the Author**

Reviewer #1: All comments have been addressed

Reviewer #3: All comments have been addressed

2. Is the manuscript technically sound, and do the data support the conclusions?

Reviewer #1: Yes

Reviewer #3: Yes

3. Has the statistical analysis been performed appropriately and rigorously?

Reviewer #1: Yes

Reviewer #3: Yes

4. Have the authors made all data underlying the findings in their manuscript fully available?

Reviewer #1: Yes

Reviewer #3: No

5. Is the manuscript presented in an intelligible fashion and written in standard English?

Reviewer #1: Yes

Reviewer #3: Yes

**Reviewer #1:**  (No Response)

**Reviewer #3:**  (No Response)

**Do you want your identity to be public for this peer review?** For information about this choice, including consent withdrawal, please see our Privacy Policy

Reviewer #1: No

Reviewer #3: No

---

## [Author Response · Author response to Decision Letter 2]

5 Aug 2025

Dear Editor

Thank you for reviewing our manuscript and for the constructive comments. Below we addressed each point. We will look forward to your feedback.

Major Concerns

Comment #1: Statistical power and model reliability: The number of cases of precocious puberty (50 cases) and delayed puberty (54 cases) is too low. Especially, the exposure model includes 9 chemicals, which may lead to overfitting of the model and unstable estimates (e.g., OR for boys with precocious puberty = 2.34, 95% CI: 0.36 - 15.14). The authors mentioned that the test power is only about 60% (assuming OR = 1.5), but did not fully discuss its potential impact on negative results (such as the possibility that no association with other chemicals may be due to insufficient power). I think the discussion section needs to be expanded to clearly explain the impact of small sample size on the mixed model (such as the risk of overfitting) and suggest the required sample size for future studies. Add sensitivity analysis and add model stability tests (such as Bootstrap method to verify the confidence interval of OR).

Response #1: We appreciate the editor’s comments. We have expanded the Discussion section to acknowledge the potential risk of overfitting and unstable estimates in our nine-chemical mixture model (Lines 301-317). However, given the limited research on the association between multiple chemical exposures and early puberty, our exploratory analysis may serve as a starting point to stimulate further investigation in this area (Lines 317-319, p.17).

Comment #2: Limitations of exposure assessment: Relying solely on a single biological sample (blood/urine) cannot reflect the dynamic of long-term exposure, especially for heavy metals with long half-lives (such as lead accumulation in bones). Other environmental pollutants (such as pesticides, flame retardants) were not included, which may miss important confounding factors. The authors need to strengthen the discussion and clearly state the limitations of single measurement in the "Limitations" section. It is recommended that future studies adopt a longitudinal design or cumulative exposure indicators. Supplementary analysis: If the data permit, analyze the interactions between chemical substances (such as heavy metals and phthalates).

Response #2: We clearly have addressed that single blood/urine measurements may not capture long‑term exposure and that additional pollutants were not assessed, potentially leaving residual confounding. We therefore recommended future longitudinal designs or cumulative exposure biomarkers.

To address the potential interaction by heavy metal exposure, we assessed stratified associations between mixture exposure to BPA, and phthalate metabolites and abnormal puberty by low (less than median) or high (median or higher) urinary cadmium level. The additional findings are described in our manuscript (Lines 164-167 & 226-228; S5 Figure).

Comment #3: Rigor of result interpretation: Only MBzP showed a significant effect on girls' delayed puberty (OR = 2.75), while the other associations were not significant or had wide confidence intervals. However, the conclusion that "chemical mixtures are associated with delayed puberty" may be an overinterpretation. The discussion on the mechanism of urban-rural differences (with a higher proportion of girls experiencing delayed puberty in rural areas) is insufficient (such as whether it is related to pollutant exposure levels or social economic factors?). Please revise the conclusion statement to avoid suggesting a "wide correlation of mixtures", and focus on the significance of MBzP and the potential role of lead in boys. Delve deeper into the differences between urban and rural areas, and supplement with literature support (such as citing Oh et al. 2024's discussion on air pollution and precocious puberty). Distinguish the effects of pollution exposure from those of confounding factors such as obesity.

Response #3: We have revised our discussion to focus on the robust MBzP association with girls’ delayed puberty and the suggestive lead effect in boys, removing any language implying a positive mixture effect (Line 241-243, p.14). The discussion in our revised manuscript explores urban–rural differences in more depth, with additional citation of similar findings (Line 287-289, p.15-16). According to our conceptual pathway explaining the association between chemical mixture exposure and puberty timing, urban residence and obesity could be separate confounders.

Comment #4: Methodological details and data accuracy: In Table 3, "MBaP" should be "MBzP" (Page 21). Inconsistent ethical approval numbers: The main text states 2013-05-004-001 (Page 12), but the supplementary material is 2013-05-004_(III).pdf (Page 38). The correlation coefficients are extremely high (such as MEOHP-MEHHP, ρ=0.97) and the impact of multicollinearity on the model has not been discussed. The authors should correct the data errors, unify the ethical approval numbers, and correct "MBaP" in Table 3. Provide an explanation for handling multicollinearity, clearly explain how QGC reduces the influence of multicollinearity (such as quantile transformation), or provide a VIF analysis in the supplementary material.

Response #4: We corrected typos in the manuscript. The ethical‑approval number is consistently listed as 2013‑05‑004‑001. You can check the same number at the ethical approval report by opening the file. However, the number shown in the file name in the supplementary data is missing the trailing "001". Compared to most traditional multivariable linear regression models that suffer from collinearity issues when including multiple correlated variables, QGC method can reduce the impact of extreme correlations between individual exposures by working with quantiles of mixtures rather than the original continuous values (Lines 156-158, p.7). In addition, QGC estimates a single overall effect (psi) for a simultaneous one-quantile increase in all exposures, rather than separate β-coefficients for each original exposure. Because we do not interpret or report separate parameter estimates that could be inflated by collinearity, the VIF is neither meaningful nor necessary (Keil AP, Buckley JP, O'Brien KM, Ferguson KK, Zhao S, White AJ. A Quantile-Based g-Computation Approach to Addressing the Effects of Exposure Mixtures. Environ Health Perspect. 2020;128(4):47004.).

Minor Concerns

Comment #1: Chart optimization: Table 1: Add units (such as blood lead μg/dL) and provide a footnote stating "median [IQR]" or "mean (SD)". Figure S4: Simplify the correlation coefficient graph, highlighting the high correlations with |ρ| > 0.7.

Response #1: In Table 1 we added units (e.g., blood lead μg/dL; urinary phthalates μg/g creatinine) and footnotes indicating “median [IQR]”(Table 1). Figure S4 has been revised to highlight only correlations |ρ|>0.7 (S4 Figure).

Comment #2: Discussion Extension: Analyze the potential mechanisms underlying the significant effect of MBzP in girls and the lack of such effect in boys (such as differences in sex hormone pathways). Refer to the latest literature to support the discussion on the urban-rural disparity.

Reponse #2: We added discussion of why the association between MBzP and abnormal puberty was not evident in boys of our study population. Because the positive association between phthalate exposure and delayed puberty was stronger in obese boys. This may explain the weaker association among the boys of our study population who show generally low BMI. We additionally cited a recent related literature (Lines 278-280, p.15).

Comment #3: Terminology Consistency: The term "phthalate metabolites" (instead of "phthalates") is uniformly used throughout the text to avoid confusion between the parent compound and its metabolites.

Response #3: We have reviewed the paper thoroughly and ensured all instances of “phthalates” mean the exposure, not the urinary biomarkers, to clarify we measured metabolites rather than parent compounds.

---

## [Decision Letter · Decision Letter 2]

29 Oct 2025

Dear Dr. Ha,

Thank you for submitting your manuscript to PLOS ONE. After careful consideration, we feel that it has merit but does not fully meet PLOS ONE’s publication criteria as it currently stands. Therefore, we invite you to submit a revised version of the manuscript that addresses the points raised during the review process.

We look forward to receiving your revised manuscript.

Kind regards,

Li Yang, M.D.

Academic Editor

PLOS ONE

Journal Requirements:

Additional Editor Comments:

Thanks for submitting your revised paper to PLOS ONE. Your manuscript has now been assessed by our editorial team and external peer experts. Although the study is of potential significance, some additional problems were proposed. Please further address the reviewers' concerns.

Reviewer's Responses to Questions

**Comments to the Author**

Reviewer #4: All comments have been addressed

Reviewer #5: All comments have been addressed

Reviewer #6: (No Response)

Reviewer #7: (No Response)

2. Is the manuscript technically sound, and do the data support the conclusions?

Reviewer #4: Partly

Reviewer #5: Partly

Reviewer #6: Yes

Reviewer #7: Yes

3. Has the statistical analysis been performed appropriately and rigorously?

Reviewer #4: No

Reviewer #5: Yes

Reviewer #6: Yes

Reviewer #7: Yes

4. Have the authors made all data underlying the findings in their manuscript fully available?

Reviewer #4: Yes

Reviewer #5: Yes

Reviewer #6: Yes

Reviewer #7: Yes

5. Is the manuscript presented in an intelligible fashion and written in standard English?

Reviewer #4: Yes

Reviewer #5: (No Response)

Reviewer #6: (No Response)

Reviewer #7: Yes

Reviewer #4: The present study includes an overall adequate sample size and has applied sound analytical methods. However, the way results are presented, together with certain limitations—some of which were already acknowledged in the manuscript—makes the findings less interpretable and difficult to draw valid inferences from:

The sample size in the groups with precocious or delayed puberty is very small, especially when analyses are stratified by sex. There were 50 children with precocious puberty (38 girls and 12 boys) and 54 children with delayed puberty, of whom only 12 were girls. This limited subgroup size substantially reduces the robustness of the statistical analyses.

Regarding the inclusion of “having siblings” in the DAG, the rationale for its relationship with the level of exposure to heavy metals, is unclear. Further clarification is needed on how this variable may confound or mediate the associations under study.

Among boys, because pubertal status was defined based on the current questionnaire, boys aged 9–13 years were excluded from the study. However, a similar exclusion was not applied to the girls. The justification for this difference between sexes is not provided.

The manuscript reports that 874 girls were included in the study, yet Table 1 indicates that 771 girls were included in the total cohort and 387 in the delayed puberty cohort. This discrepancy needs to be clarified.

In lines 184–188, the authors make comparisons that are statistically invalid. For example, they state that more girls were in the precocious puberty group (38 of 50). Since the number of boys and girls in the cohort is not equal, such a comparison is not appropriate. Instead, the correct approach would be to compare the proportion of children experiencing precocious puberty within each sex cohort (i.e., the percentage of girls versus the percentage of boys)

The definition of precocious and delayed puberty in your study is inherently age-dependent, as early puberty is only detectable in younger children and delayed puberty only in older children. Consequently, age is strongly collinear with the outcome. Entering age directly into the regression model may therefore lead to problems such as overadjustment or complete separation, making the estimates unstable or difficult to interpret. A more appropriate strategy would be to restrict the analysis to an age window where both exposed and unexposed children with and without the outcome are observed or..

Reviewer #5: "Our findings contribute to the growing body of evidence implicating environmental chemicals that have

lifelong health impacts on pubertal development."

The current study did not focus on all the environmental chemicals so it should be recommended not to say environmental chemicals rather sharing in a more specific way regarding heavy metals and other chemicals included in the study. Longitudinal study should be needed to say the lifelong impact.

Reviewer #6: The author has addressed the reviewers' comments, implemented appropriate revisions, and incorporated additional content that enhances the overall completeness and scholarly rigor of the manuscript. The study demonstrates a degree of innovation and holds academic significance within the relevant field. It is therefore recommended for acceptance.

Reviewer #7: Using data from the Korea Environmental Exposure and Health Survey in Children and Adolescents, the authors conducted a cross-sectional study to look at the associations between exposure to heavy metals, bisphenol A, and phthalates and precocious or delayed puberty. The authors reported that a higher mono-benzyl phthalate concentration was associated with a higher risk of delayed puberty in girls in single-exposure models, concluding that exposure to environmental chemicals may contribute to delayed puberty.

Overall, if the research questions examined in this study were to be replicated in numerous robust studies, the findings could inform public health interventions aimed at reducing the impact of modifiable environmental pollutants on puberty.

With that in mind, this reviewer has the following to remark:

1. The authors have made substantial modifications to their manuscript, improving clarity and overall quality. This is evident from the detailed feedback provided by the editor and reviewers.

2. However, a major concern with this study is statistical power. When interpreting the results, it’s crucial to assess whether statistical power was sufficient to detect at least moderate-effect sizes to answer the research questions raised.

While the authors, as recommended by the editor, have acknowledged the potential risks of overfitting and unstable estimates in their multi-exposure model, the paper could guide future studies on how to proceed. In their discussion section, the authors could suggest strategies, apart from increasing the sample size (which is a given), to improve the likelihood of detecting a true effect, if one exists.

I hope this review is helpful and wish the authors the very best with their research!

**Do you want your identity to be public for this peer review?** For information about this choice, including consent withdrawal, please see our Privacy Policy

Reviewer #4: **Yes: ** Mohammad Aghaali

Reviewer #5: **Yes: ** Muhammed Ashraful Alam

Reviewer #6: No

Reviewer #7: **Yes: ** Dr. Widad Akreyi

---

## [Author Response · Author response to Decision Letter 3]

8 Nov 2025

Dear Editor

We appreciate the reviewer’s comments. Please find our point-by-point responses and revised manuscript. We will look forward to your feedback.

Best regards,

<Reviewer #4>

Comment #1: The present study includes an overall adequate sample size and has applied sound analytical methods. However, the way results are presented, together with certain limitations—some of which were already acknowledged in the manuscript—makes the findings less interpretable and difficult to draw valid inferences from:

The sample size in the groups with precocious or delayed puberty is very small, especially when analyses are stratified by sex. There were 50 children with precocious puberty (38 girls and 12 boys) and 54 children with delayed puberty, of whom only 12 were girls. This limited subgroup size substantially reduces the robustness of the statistical analyses.

Response #1: We appreciate the reviewer’s comments. We agree that the subgroup sizes were small, especially for boys with precocious puberty and girls with delayed puberty. We have addressed an explicit acknowledgment of limited power (about 60%), and emphasized that the study is exploratory, aiming to provide effect-size estimates to inform future longitudinal or pooled analyses (Lines 300-301; 304-305, p.16; 324-326, p.17 in the revised manuscript).

Comment #2: Regarding the inclusion of “having siblings” in the DAG, the rationale for its relationship with the level of exposure to heavy metals, is unclear. Further clarification is needed on how this variable may confound or mediate the associations under study.

Response #2: Having siblings can reflect differences in household characteristics that may indirectly influence both exposure levels and developmental outcomes. Including this variable helps to partially control unmeasured household-level confounding (Lines 140-142, p.6-7).

Comment #3: Among boys, because pubertal status was defined based on the current questionnaire, boys aged 9–13 years were excluded from the study. However, a similar exclusion was not applied to the girls. The justification for this difference between sexes is not provided.

Response #3: The exclusion of boys aged 9–13 years was based on the definition of precocious and delayed puberty using Tanner stage criteria. While girls in this age group were eligible for precocious puberty identification, determining precocious or delayed puberty for boys aged 9–13 years was not possible based on the single measurement taken at the time of the survey. This issue was addressed in Method (Line 130-132, p.6).

Comment #4: The manuscript reports that 874 girls were included in the study, yet Table 1 indicates that 771 girls were included in the total cohort and 387 in the delayed puberty cohort. This discrepancy needs to be clarified.

Response #4: Thanks to the reviewer, we corrected the number in Table 1. The total number of girls included in the analysis of precocious puberty was 874 (Table 1).

Comment #5: In lines 184–188, the authors make comparisons that are statistically invalid. For example, they state that more girls were in the precocious puberty group (38 of 50). Since the number of boys and girls in the cohort is not equal, such a comparison is not appropriate. Instead, the correct approach would be to compare the proportion of children experiencing precocious puberty within each sex cohort (i.e., the percentage of girls versus the percentage of boys)

Response #5: We agree with the reviewer and have revised the text, presenting the percentage of girls versus the percentage of boys (Lines 189-191, p.9).

Comment #6: The definition of precocious and delayed puberty in your study is inherently age-dependent, as early puberty is only detectable in younger children and delayed puberty only in older children. Consequently, age is strongly collinear with the outcome. Entering age directly into the regression model may therefore lead to problems such as overadjustment or complete separation, making the estimates unstable or difficult to interpret. A more appropriate strategy would be to restrict the analysis to an age window where both exposed and unexposed children with and without the outcome are observed or..

Response #6: We have clarified that age was included as a covariate because it can act as a common cause influencing both chemical exposure and pubertal status. Excluding age from the model would have introduced residual confounding, although the potential multicollinearity between the covariates. Given that pubertal status was assessed using Tanner staging, it was not feasible to apply age-standardized measures in our study (Lines 139-140, p.6).

<Reviewer #5>

Comment #1: Effects of background factors and other issues were difficult to control even in single- and joint-exposure analyses performed in the current study. It’s another limitation which should be clarified. Actually there are so many limitations in the current study.

Response #1: We agree that unmeasured or residual confounding attributed to background factors is an important limitation. Unmeasured environmental, nutritional, or genetic factors may remain as a source of residual confounding effects. We have clarified these limitations in the revised manuscript to ensure balanced interpretation of our findings (Lines 306-310, p.16).

Comment #2: "Our findings contribute to the growing body of evidence implicating environmental chemicals that have lifelong health impacts on pubertal development."

The current study did not focus on all the environmental chemicals so it should be recommended not to say environmental chemicals rather sharing in a more specific way regarding heavy metals and other chemicals included in the study. Longitudinal study should be needed to say the lifelong impact.

Response #2: We appreciate the reviewer’s comment. The conclusion has been revised to specify the chemical classes studied: “Our findings contribute to the growing body of evidence implicating the potential role of heavy metals, BPA and phthalate metabolites in pubertal development.” (Lines 335-336, p.17).

<Reviewer #6>

Comment: The author has addressed the reviewers' comments, implemented appropriate revisions, and incorporated additional content that enhances the overall completeness and scholarly rigor of the manuscript. The study demonstrates a degree of innovation and holds academic significance within the relevant field. It is therefore recommended for acceptance.

Response: We appreciate the reviewer’s comment.

<Reviewer #7>

Using data from the Korea Environmental Exposure and Health Survey in Children and Adolescents, the authors conducted a cross-sectional study to look at the associations between exposure to heavy metals, bisphenol A, and phthalates and precocious or delayed puberty. The authors reported that a higher mono-benzyl phthalate concentration was associated with a higher risk of delayed puberty in girls in single-exposure models, concluding that exposure to environmental chemicals may contribute to delayed puberty.

Overall, if the research questions examined in this study were to be replicated in numerous robust studies, the findings could inform public health interventions aimed at reducing the impact of modifiable environmental pollutants on puberty.

With that in mind, this reviewer has the following to remark:

Comment #1: The authors have made substantial modifications to their manuscript, improving clarity and overall quality. This is evident from the detailed feedback provided by the editor and reviewers.

Response #1: We appreciate the reviewer’s comment.

Comment #2: However, a major concern with this study is statistical power. When interpreting the results, it’s crucial to assess whether statistical power was sufficient to detect at least moderate-effect sizes to answer the research questions raised.

While the authors, as recommended by the editor, have acknowledged the potential risks of overfitting and unstable estimates in their multi-exposure model, the paper could guide future studies on how to proceed. In their discussion section, the authors could suggest strategies, apart from increasing the sample size (which is a given), to improve the likelihood of detecting a true effect, if one exists.

I hope this review is helpful and wish the authors the very best with their research!

Response #2: We agree that the subgroup sizes were small, especially for boys with precocious puberty and girls with delayed puberty. We have explicitly acknowledged the limited power (60%) of our study in Discussion. We emphasized that the study is exploratory, aiming to provide effect-size estimates to inform future longitudinal or pooled analyses (Lines 300-301; 303-305, p.16).

---

## [Decision Letter · Decision Letter 3]

9 Dec 2025

Exposure to heavy metals, bisphenol A, and phthalates: Implications for precocious or delayed puberty

PONE-D-25-12742R3

Dear Dr. Ha,

We’re pleased to inform you that your manuscript has been judged scientifically suitable for publication and will be formally accepted for publication once it meets all outstanding technical requirements.

Kind regards,

Li Yang, M.D.

Academic Editor

PLOS One

Additional Editor Comments (optional):

Thanks for the authors' efforts to comprehensively improve your manuscript according to editor's and reviewers' comments. I am pleased to inform you that your paper can be accepted for publication now. Thanks for the chance to assess your important work. Additionally, many thanks for all the reviewers' precious inputs.

Reviewers' comments:

Reviewer's Responses to Questions

**Comments to the Author**

Reviewer #4: All comments have been addressed

Reviewer #7: (No Response)

2. Is the manuscript technically sound, and do the data support the conclusions?

Reviewer #4: Yes

Reviewer #7: (No Response)

3. Has the statistical analysis been performed appropriately and rigorously?

Reviewer #4: Yes

Reviewer #7: (No Response)

4. Have the authors made all data underlying the findings in their manuscript fully available?

Reviewer #4: Yes

Reviewer #7: (No Response)

5. Is the manuscript presented in an intelligible fashion and written in standard English?

Reviewer #4: Yes

Reviewer #7: (No Response)

Reviewer #4: (No Response)

Reviewer #7: (No Response)

**Do you want your identity to be public for this peer review?** For information about this choice, including consent withdrawal, please see our Privacy Policy

Reviewer #4: **Yes: ** Mohammad Aghaali

Reviewer #7: **Yes: ** Dr. Widad Akreyi

---

## [Editor Report · Acceptance letter]

PONE-D-25-12742R3

PLOS One

Dear Dr. Ha,

I'm pleased to inform you that your manuscript has been deemed suitable for publication in PLOS One. Congratulations! Your manuscript is now being handed over to our production team.

Kind regards,

on behalf of

Dr. Li Yang

Academic Editor

PLOS One